# Taxonomy of Factors Involved in Decision-Making to Sustain Organization Members' Creativity

**Martina Blašková** [1] , **Dominika Tumová** [2] **and Martin Mičiak** [2,*]

1. Department of Management and Informatics, Police Academy of the Czech Republic in Prague, 143 01 Prague, Czech Republic; blaskova.fri@gmail.com
2. Department of Managerial Theories, University of Žilina, 010 26 Žilina, Slovakia; dominika.tumova@uniza.sk
* Correspondence: martin.miciak@uniza.sk

**Abstract:** Creativity is highly valued in all areas of life, and it must be supported in the academic environment for the future prosperity of all organizations. This is the primary source of creativity for practice. The research presented is based on answering the following questions: *Are an organization's members willing to increase their motivation if the organization's motivational efforts improve? What elements in decision-making are used to support the creativity and motivation of the organization's members?* The paper proposes a novel taxonomy of decision-making factors influencing organizations' sustainable creativity based on linking the findings from the authors' surveys. Its application will lead to an improvement in the organizations' processes, especially in the process of education and knowledge generation. The sociological inquiry was used as the main data collection method. Other methods applied included content analysis, practical cases analysis, and comparison. Methods of mathematical-statistical analysis and deductive-inductive approach were used in the evaluation. This resulted in the confirmation of the impact of creative decision-making approaches by employees and managers on sustainable motivation and creativity in the organization. The taxonomy of decision-making factors for the support of sustainable creativity reflects the results of this original research combined with the findings summarized in the discussion.

**Keywords:** organization behavior; learning and development; decision-making factors; motivation; sustainable creativity; higher education

## 1. Introduction

It is higher education that plays a key role in the overall process of striving for quality and sustainable development. Universities act as a role model, contribute via societal outreach (Barth 2021), and enable students to achieve motivation, creativity, and quality of their own processes and of overall mechanisms in the state. HR professionals in higher education must prepare and constantly refine learning environments to build the conditions for sustainable quality.

This paper approaches the quality of working and studying conditions (Ibragimova and Bagaeva 2018) by examining three processes that take place continuously in the academic environment. These include *motivation, creativity, and decision-making aimed at supporting them.* Although these processes are partially researched (e.g., Marques 2016; Ryan and Deci 2017; Averill and Major 2020; Kleebbua and Lindratanasirikul 2021), the examination of their overlaps, specifically in higher education, is still absent in the literature. The paper will identify the interconnections between all three processes while accepting their unique features.

Motivation is an inner state that energizes and sustains behavior towards a goal (Khanal et al. 2021, p. 83). It is crucial to support the motivation of university teachers because according to Sinclair (2008), if teachers are to be sustained in teaching for a long time, they must be satisfied. Then they can be expected to teach with enthusiasm and

dedication. From an analogous perspective, it is essential to constantly support motivation of all other actors in the academic environment (students, managers, researchers, etc.).

Creativity and its sustainability are important features for the whole society, with creativity considered as a self-actualizing process, fulfilling basic human needs (d'Orville 2019, p. 66). Within the academic environment, creativity is a necessary component contributing to the creation and transformation of original ideas.

Another dimension of the analyzed phenomena interconnectedness is contained in the assumption of using the teachers' ability to motivate students to be creative. This ability is conditioned by the motivation of teachers towards such activities. If teachers are motivated to support the creativity of their students, a sustainably high level of their creativity can be achieved (Agnoli et al. 2018).

Additionally, the teachers' willingness to motivate students to be creative must not be absent. To achieve sustainable support for creativity, teachers must truly use creative approaches in decisions about supporting both students' motivation and creativity. Teachers will be able to encourage students to have a sustainable level of creativity supported by their intrinsic motivation, and HR professionals will be able to intensify the same processes among teachers. The fusion of these elements represents the achievement of increased educational process quality (Guney and Al 2012) and improved conditions of work and study. The paper presents concentrated results of in-depth research on the relationship between creativity and sustainability (Figure 1).

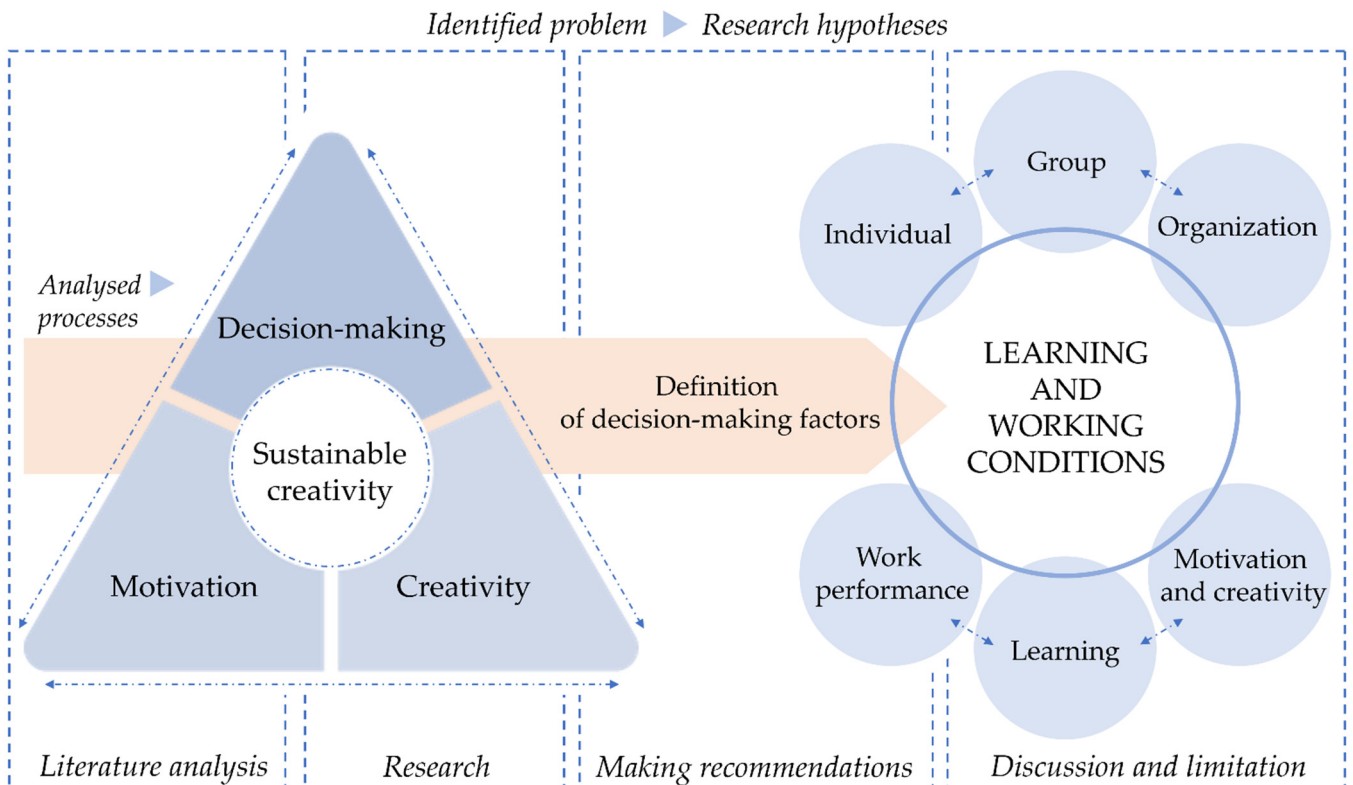

**Figure 1.** Mutual links between the research areas.

The aim of the paper is linked to the creation of the taxonomy of decision-making factors influencing the building of sustainable creativity in organizations for their development. Following the aim, the researched question's formulation is: *Is it possible to influence the building of sustainable motivation and sustainable creativity by the appropriately set decision-making processes?*

## 1.1. Decision-Making on Sustainable Creativity

Due to learning and motivation, it is possible to direct people's behavior to seek new knowledge (Koman et al. 2018) so that personal improvement and sustainable creativity are achieved (Carrera and Ramírez-Hernández 2018). This idea characterizes the interconnection of three basic areas (motivation, creativity, and decision-making), which the authors decided to explore with an emphasis on building sustainable creativity (Tumová and Blašková 2021).

### 1.1.1. Sustainable Motivation

The organization's sustainability can be described as the ability to adapt to changes and create a range of opportunities for providing effective services (Varmus et al. 2018). Another view of sustainability is supported by Schalock et al. who states that it is a multidimensional phenomenon focused on maintaining good results, generating knowledge, or support based on values (Schalock et al. 2016).

Motivation can be considered an internal process associated with several outcomes: curiosity, perseverance, willingness to learn, etc. (Vallerand et al. 1992). In this paper, motivation is viewed through the behavior of individuals and groups. Manifestations of human behavior in relation to process theory are revealed by McInerney, who argues that motivation is based on an inner desire to improve and understand phenomena and their interrelationships (McInerney 2005). It is possible to draw attention to mental processes, such as perception or one's adaptation to a situational context (Oyserman and Destin 2010; Oyserman 2013). If employees and managers are willing to shape their motivation permanently, it becomes sustainable (Blašková et al. 2018).

Finke and Will defined several measures leading to the activation of sustainable motivation: communication management, competence development, adaptation of management systems, and integration (Finke and Will 2003). One of the theories enabling a better understanding of factors influencing the motivation of individuals and groups is the self-determination theory (Ryan and Deci 2017).

The authors' assumed the existence of an internal interconnection of areas of creativity and decision-making for effective support of the academic environment members' motivation. If sustainable motivation (Tsoi et al. 2018; Yarmakeev et al. 2019) of the organization's members is to be achieved in the long run, creative approaches must be used during planning. When creativity is used in managerial decision-making, it becomes creative itself. This leads to the formulation of Hypothesis 1.

**Hypothesis 1 (H1).** *The utilization of creative decision-making approaches, methods, and procedures by employees and managers supports the sustainable motivation of the organization's members.*

The research's orientation on the motivation of the academic environment members is based on the fact that this is a fundamental building block of achieving the final quality of the educational process.

### 1.1.2. Sustainable Creativity

Creativity can be understood as a mental process developing new ideas (Branscomb and Auerswald 2002). The interconnectedness of the concepts of creativity and sustainability is highlighted by several authors' ideas, such as 'Creativity and sustainability are closely linked' (d'Orville 2019). Attention can be paid to how creativity could help achieve environmental sustainability. The International Center for Creativity and Sustainable Development also focuses on this. It can be considered an international think tank for creativity development, standing behind the initiative CREATIVITY 2030 (UNESCO 2020). Via creativity, it is possible to contribute to a more sustainable world (Marques 2016). Another perspective, researched in this article, is how to encourage creativity in the long run so that it becomes sustainable itself. There has been a growing interest in this connection recently (Craze 2016).

Defining creativity is complex, as Blamires and Peterson point out. They concluded that creativity is 'an individual psychological property' and 'development of collaborative endeavor' (Blamires and Peterson 2014). It represents not only individual quality but also a process that brings original ideas with an emphasis on the importance of creative collaboration. Also, attention must be paid to how the process of creativity can become sustainable. Boyatzis characterized the term Ideal Creative Self as developing and motivating the core that reflects hopes, dreams, and goals (Boyatzis 2006). Cognitive and affective processes are involved in influencing the ideal self, forming motivation and creativity. If people work with their motivation and creativity, they can use emotional intelligence to assess the conformity of the desired future state with basic values (Gilbert and Wilson 2007).

Working with creativity can be understood as a learning process. One of the main purposes of education should be to explain the meaning of the phenomenon given. Only then will the motivation of those involved in teaching be fully utilized to create new solutions (Carrera and Ramírez-Hernández 2018; Clegg and Burdon 2021). Creativity can be seen as an indicator of learning effectiveness in the educational process (Shu et al. 2020).

Promotion of creativity and evaluation of its support have been analyzed by many authors (e.g., Gabriel et al. 2016; Lee et al. 2016). It is necessary to examine the specific possibilities for this support. Richardson and Mishra defined concrete elements for the support of students' creativity: learner engagement, physical environment, and learning climate (Richardson and Mishra 2018).

The focus of the H2 hypothesis was specified as a direct influence of creative approaches in decision-making to support academic environment members' sustainable creativity (Terzidis and Darbellay 2017; Svejdarova 2020). The increased creativity is also an indicator of the teaching process's effectiveness (Shu et al. 2020). Therefore, the H2 hypothesis was formulated as follows:

**Hypothesis 2 (H2).** *The utilization of creative decision-making approaches, methods, and procedures by employees and managers supports the sustainable creativity of the organization's members.*

The importance of exploring these influences to support creativity is derived from the fact that if high creativity is achieved in the learning environment, the whole environment becomes creative.

### 1.1.3. Decision-Making to Sustain Creativity

Many authors emphasize the importance of decision-making for supporting creativity in the educational process. The specific students' attributes were revealed, which improved after the application of the pre-selected approach by the managers and the teachers. The precondition is a positive impact of applying specific measures in the decision-making process to support creativity (Vallerand et al. 1992; Doyle 1979; Criss 2011; Powell 2007; Browman and Destin 2016; Soares and Lopes 2020).

One of the conceptual approaches in which the decision-making process to support creativity can be embedded is the EFQM model of excellence. This model can be characterized as 'creating a sustainable future' (CGMA 2018). In this context, sustainable creativity is a fundamental prerequisite for the gradual building of an excellent organization. Such organizations have a positive impact on various stakeholders (Jankalová and Jankal 2017).

One of the presumptions for improving the quality of the decision-making process is the application of creative procedures. The above is closely linked to the formulation of the H1 and H2 hypotheses. However, the H3 hypothesis broadens the perspective of decision-making (Huang and Wang 2021; Kahn and Bullis 2021) related to the promotion of motivation and creativity. It focuses on the overall improvement of the decision-making process quality.

**Hypothesis 3 (H3).** *Improving the quality of the decision-making process on the development of motivation and creativity has the effect of improving the creativity and motivation of the organization's members.*

The achieved effect of increased motivation and creativity of the academic environment members will significantly contribute to the increased quality of education. The action of these mutual forces will have a synergistic effect. Motivated and creative managers and teachers set up a decision-making process that creates a motivating and creative learning environment.

1.1.4. Higher Education Organizations as Personnel Systems, with Motivation and Creativity Accent

Higher education quality is currently being disputed more often than before. Especially, in the face of growing societal challenges, international scientific cooperation is needed (Soviar et al. 2017). It is the concept of quality that gives higher education its status (Gupta 2021). The quality must pervade all of a university's organizational units (rector's office, faculties, departments), members (teachers, scientists, managers), and activities (educational, scientific, publishing, developmental, etc.). In addition, the quality of decision-making in the field of motivation, creativity, and sustainability is critical for the establishment of positive relationships (Wright and Silard 2021) and climate in organizations.

Universities generate and transfer relevant knowledge, as well as enable future change agents to contribute to a sustainable future (Barth 2021). The quality of higher education must be advanced as it applies to the whole culture of the country (Dhal 2013) and the world. Nesterova et al. (2019) and Ortiz-Herrera et al. (2020) underline that universities must reorient their teaching, research, networking, and management activities so that they are permanently being performed with high quality.

Cheng argues for a motivationally intelligent quality approach, emphasizing the moral dimension and the intrinsic values of academics and students (Cheng 2016). Sustainable motivation needs to be supported creatively and responsibly. University bodies must respect a myriad of motives, powers, ambitions, and contradictions. Here, trust is a fundamental phenomenon of both motivation and creativity (Nesterova et al. 2020). It captures overall motivational dimensions, i.e., the intra-motivational and inter-motivational ones, and 'dynamizes the future quality of the processes' (Hultman 2002, p. 40).

Furthermore, universities must create 'the most effective package of monetary and non-monetary rewards for motivating performance, attracting and retaining talent, and maintaining employee morale' (Ledford 2003) because job motivation positively affects teachers' satisfaction (Arifin 2015). Teachers must be strongly committed to making the change in their practice to sustain their effort (Zuzeviciute et al. 2016; Averill and Major 2020).

Studies show that motivational systems are usually very effective after their implementation and elicit the desired motivation (Sale 2016; Bylkov and Har'kina 2021). However, as human beings are dynamic (Krushelnitskaya et al. 2019), the effectiveness of motivational systems decreases over time. Therefore, it is necessary not only to motivate the university staff and students, but to do so creatively.

## 2. Results

Relevant relationships between the variables were tested to identify statistically significant ones. These form the basis for verifying the validity of research hypotheses.

New variables were also created based on: (1) assigning points to the options; (2) recalculating the average level of a multi-question elements; and (3) combining the options of a particular question for comparison to obtain interpretable results for testing hypotheses.

### 2.1. Hypothesis 1

To test the validity of Hypothesis 1, the relationships between the selected questions were examined (Table 1). Creative decision-making approaches consisted of five selected elements (1–5). The aspect of sustainable motivation was examined in the branch of the current level of the respondents' average motivation, as well as efforts to improve oneself in the future. Most relationships proved to be statistically significant, thus their specific frequencies were further analyzed.

**Table 1.** Statistical significance of the relationship between selected questions and sustainable motivation—teachers' and students' perspective.

| Analyzed Questions | | Average Level of Overall Motivation | | Willingness to Increase the Level of Effort and Motivation | |
|---|---|---|---|---|---|
| | | Employees | Students | Employees | Students |
| (1) Perceiving the approach: participatory, neutral, authoritative | Chi-Square/Z-Score C *P*-value Significance | χ2 (3) = 8.514 7.815 0.037 yes | χ2 (3) = 43.384 7.815 <0.001 yes | z = 3.834 1.96 <0.001 yes | z = 1.453 1.96 1.46 no |
| (2) Perceiving open communication | Chi-Square C *P*-value Significance | χ2 (12) = 44.951 21.026 <0.001 yes | χ2 (9) = 71.358 16.919 <0.001 yes | χ2 (4) = 17.844 9.488 0.001 yes | χ2 (3) = 16.251 7.815 0.001 yes |
| (3) Perceiving an environment of trust, helpfulness, and belonging | Chi-Square C *P*-value Significance | χ2 (12) = 19.764 21.026 0.072 no | χ2 (12) = 169.622 21.026 <0.001 yes | χ2 (4) = 19.824 9.488 <0.001 yes | χ2 (4) = 13.524 9.488 0.009 yes |
| (4) Perceiving motivation to be creative | Chi-Square C *P*-value Significance | χ2 (9) = 37.71 16.919 <0.001 yes | χ2 (12) = 77.3 21.026 <0.001 yes | χ2 (3) = 9.075 7.815 0.028 yes | χ2 (4) = 5.624 C = 9.488 0.229 no |
| (5) Perceiving creative ideas recognition | Chi-Square C *P*-value Significance | χ2 (12) = 21.096 21.026 0.049 yes | χ2 (12) = 168.197 21.026 <0.001 yes | χ2 (4) = 14.876 9.488 0.005 yes | χ2 (4) = 1.58 9.488 0.821 no |

The mechanism for calculating the average level of motivation consisted of assigning points to individual answers (highest level = 5 points; lowest level = 1 point), adding values for specific areas of motivation, dividing the result by the number of areas, and assigning new intervals.

Based on Table 1, some differences between the current and future level of motivation can be seen in the students' opinions. A total of 94% of respondents who perceived that a participatory approach (1) was being applied towards them have a high or very high level of motivation. Only 6% of the respondents (*n* = 244) had a low or rather lower level of current motivation. However, regarding the future, the statistical significance of the relationship examined was not confirmed.

The remaining four elements of creative decision-making approaches were examined via relevant questions, where respondents could express their views on the application of the factors examined using a five-point scale (1 = yes; 2 = mostly yes; 3 = sometimes yes; 4 = mostly no; 5 = no). An example is factor (2), where 90% of respondents who perceived the environment of open communication had a very high or high level of motivation.

Regarding the future aspect, those respondents who perceived open communication in the environment were trying to improve their level of motivation (88%). All other statistically significant relationships were analyzed the same way. The results confirmed the assumption of a positive impact of the application of creative decision-making approaches

on the support of sustainable motivation. Statistical significance was not detected among students only for factors (4) and (5).

For employees, all relationships were characterized as statistically significant, except for factor (3) and its impact on current motivation. Studying the present state, respondents (97%) who perceived the participatory approach also had a very high or high average level of current motivation.

Another example of the influence on the present is factor (2). A total of 99% of those who perceived communication as open had a high or very high level of motivation. The relationships between the other elements of creative decision-making approaches and the current level of motivation were analyzed the same way. The findings suggest a positive impact of their application. Regarding the future aspect, it was revealed that only about 40% of respondents were willing to increase their future efforts if the five elements of creative decision-making approaches are applied.

Statistically significant relationships were not confirmed mostly regarding the willingness to increase future efforts. This does not contradict the defined hypothesis, because if a high level of motivation is already achieved, it is not possible to constantly increase it. It is appropriate to maintain motivation at a high level, as it was supported by linking two perspectives on motivators. This was a matter of linking the perceived effectiveness of the factors influencing motivation with their desired application in the future. The statistically significant relationships related to the effort to increase motivation of respondents in the future are therefore only a complementary aspect. It points to the fact that with the right choice of creative decision-making approaches, it is possible to achieve better future results. Based on these arguments, the validity of the H1 hypothesis was confirmed.

### 2.2. Hypothesis 2

This section presents the results related to the validation of the H2 hypothesis. Five aspects of creative decision-making approaches were linked to variables representing sustainable creativity: *the degree of motivation to submit new ideas and increase the effectiveness of the educational process; and the degree of motivation to creative collaboration* (Table 2).

Within the twenty relationships examined (Table 2), only four were statistically insignificant. For significant relationships, the direction of the influence was analyzed. In all the cases, the perception was studied versus the imperception of the element in relation to the high level of two specific components of motivation reflecting sustainable creativity.

All relationships were evaluated focusing on frequencies using a similar mechanism. An example is the relationship between the perception of a participatory approach (1) and the degree of motivation to submit new ideas or increasing the effectiveness of the educational process. A total of 28% of students ($n = 419$) who perceived the utilization of a participatory approach felt a high degree of motivation to submit new ideas. In contrast, only 12% felt a high degree of this type of motivation while not perceiving the application of a participatory approach.

Looking at the level of motivation to creative cooperation, up to 37% of the inquired students, who feel the application of a participatory approach (1), also feel a high level of motivation to creative cooperation.

Thus, the application of creative decision-making approaches truly influences the achievement of sustainable creativity. A closer look at these results emphasizes the importance of creative cooperation. Therefore, in promoting sustainable creativity, it is crucial to promote it.

**Table 2.** Statistical significance of the relationship between selected questions and two aspects of sustainable creativity—teachers' and students' perspective (own study).

| Analyzed Questions | | Degree of Motivation to Submit New Ideas and Increase the Effectiveness of the Educational Process | | Degree of Motivation to Creative Collaboration | |
|---|---|---|---|---|---|
| | | Employees | Students | Employees | Students |
| (1) Perceiving the approach: participatory, neutral, authoritative | Chi-Square | $\chi2 (4) = 2.213$ | $\chi2 (4) = 25.673$ | $\chi2 (4) = 18.176$ | $\chi2 (4) = 43.231$ |
| | C | 9.488 | 9.488 | 9.488 | 9.488 |
| | *P*-value | 0.697 | <0.001 | 0.001 | <0.001 |
| | Significance | no | yes | yes | yes |
| (2) Perceiving open communication | Chi-Square | $\chi2 (16) = 30.61$ | $\chi2 (12) = 17.646$ | $\chi2 (16) = 27.76$ | $\chi2 (12) = 21.917$ |
| | C | 26.296 | 21.026 | 26.296 | 21.026 |
| | *P*-value | 0.015 | 0.127 | 0.034 | 0.038 |
| | Significance | yes | no | yes | yes |
| (3) Perceiving an environment of trust, helpfulness, and belonging | Chi-Square | $\chi2 (16) = 12.706$ | $\chi2 (16) = 61.601$ | $\chi2 (16) = 40.687$ | $\chi2 (16) = 54.733$ |
| | C | 26.296 | 26.296 | 26.296 | 26.296 |
| | *P*-value | 0.694 | <0.001 | <0.001 | <0.001 |
| | Significance | no | yes | yes | yes |
| (4) Perceiving motivation to be creative | Chi-Square | $\chi2 (12) = 24.983$ | $\chi2 (16) = 61.117$ | $\chi2 (12) = 32.716$ | $\chi2 (16) = 44.093$ |
| | C | 21.026 | 26.296 | 21.026 | 26.296 |
| | *P*-value | 0.015 | <0.001 | 0.001 | <0.001 |
| | Significance | yes | yes | yes | yes |
| (5) Perceiving creative ideas recognition | Chi-Square | $\chi2 (16) = 20.796$ | $\chi2 (16) = 50.394$ | $\chi2 (16) = 28.71$ | $\chi2 (16) = 39.988$ |
| | C | 26.296 | 26.296 | 26.296 | 26.296 |
| | *P*-value | 0.186 | <0.001 | 0.026 | <0.001 |
| | Significance | no | yes | yes | yes |

*2.3. Hypothesis 3*

Based on previous research findings published (Tumová and Blašková 2021; Tumová and Demjanovičová 2021), creativity will also be supported by targeted support of motivation. Motivation and creativity are promoted by similar factors.

Therefore, only motivational factors were used to test the H3 hypothesis. These were examined regarding two aspects: *the current level of motivation and the willingness to further improve one's results.* The conclusions presented below can be related not only to motivation but also to the support of creativity (Table 3). The interconnectedness of the factors that the respondents identified as effective in influencing their motivation with those they also identified as desired in the future, represents an increase in the quality of decision-making to support motivation and creativity.

**Table 3.** Dependence between selected motivational factors and the level of teachers' and students' motivation (own study).

| Motivational Factors | | Average Level of Overall Motivation | | Willingness to Increase the Level of Effort and Motivation | |
|---|---|---|---|---|---|
| | | Employees | Students | Employees | Students |
| (1) Creating good relationships and atmosphere | Chi-Square | $\chi2 (27) = 43.167$ | $\chi2 (27) = 59.819$ | $\chi2 (9) = 4.687$ | $\chi2 (9) = 23.402$ |
| | C | 40.113 | 40.113 | 16.919 | 16.919 |
| | *P*-value | 0.025 | <0.001 | 0.861 | 0.005 |
| | Significance | yes | yes | no | yes |
| (2) Correctness from management/teachers | Chi-Square | $\chi2 (21) = 15.01$ | $\chi2 (27) = 55.793$ | $\chi2 (7) = 5.203$ | $\chi2 (9) = 32.198$ |
| | C | 32.671 | 40.113 | 14.067 | 16.919 |
| | *P*-value | 0.822 | <0.001 | 0.635 | <0.001 |
| | Significance | no | yes | no | yes |

Among students, the significance was confirmed for both examined factors, in relation to the level of current motivation and the effort to increase future motivation. An example of how the frequencies were interpreted is the perceived effectiveness of factor (1) from the students' perspective in relation to their current level of overall motivation. Within the group of respondents who perceive this factor as highly effective (options 9 or 10 on a 10-point scale), there is a considerably higher portion (91%) of those with a high current average level of motivation (options very high or rather higher). The analysis of the relationship of the factor with the willingness to improve oneself in the future brought a similar result. A total of 92% of respondents who attributed very high effectiveness to the factor are still willing to increase their efforts in the future. A detailed analysis was also performed for factor (2). The results revealed that respondents who perceived this factor as effective had a high level of current average motivation (91%) as well as the willingness to increase their efforts in the future (93%).

For employees, statistical significance was revealed only for factor (1) in relation to the current level of motivation. A total of 95% of respondents who consider this factor effective have a high or very high level of current average motivation.

Thus, the validity of the established hypothesis H3 was confirmed only for students. In their case, the statistical significance of the factors' influence was confirmed both regarding the current motivation and the effort to increase future motivation. As these factors are effective in supporting motivation as well as in promoting creativity, creativity will also be encouraged via their implementation.

## 3. Discussion

For a comprehensive view, the findings of other researchers focusing on the conditions for students and teachers were analyzed. When setting goals in supporting education, it is necessary to focus on the development of teachers' leadership and creative abilities so that creative abilities and logical thinking can be supported among students. It is also important to focus on supporting motivation to increase the level of students' results, especially via the targeted application of creative approaches (Doyle 1979). Teachers and managers should consider the use of emotions and relationships when building a pleasant learning environment (Hong and Aqui 2004; Criss 2011; Hoekman et al. 2005; Avsec and Jagiello-Kowalczyk 2021).

The results of several studies suggest that in supporting students' motivation and creativity, self-sufficiency, ability to use cognitive strategies, perception of skills, and self-efficacy, but relationships between motivation and students' optimism should be supported as well. These aspects should be developed by the teachers' targeted decision-making (Calavia et al. 2021). The role of positive emotions in the academic environment for the achievement of stable motivation and creativity reflected in the performance was emphasized by Dewaele et al. (2019) and Oriol et al. (2016).

In the past, intrapersonal and motivational factors have been considered in the design of student curricula (Clark 1988; Janos and Robinson 1985) but have often been neglected in the assessment process (Porath 1996). More recent direction in promoting creativity in education shows that ideas are transformed, innovations are formed, and implementation has a social or economic impact (Branscomb and Auerswald 2002). This direction is followed by the results of research activities performed under the auspices of the European University Association. The association launched a project called Creativity in Higher Education, where a consortium of seven European universities agreed to explore different attitudes for the development of creativity in the educational process (European University Association 2007). The fundamental elements of a creative university include: *diverse and creative leaders, creative teams, creative and flexible relationships, creative communities of professional practice,* etc. (Powell 2007; Soviar et al. 2015; Barnett 2020).

A pivotal aspect of achieving the development of motivation and creativity of an organization's members is *targeted decision-making*, which leads to building trust between the organization and its members. These aspects form the climate of the educational

institution (Browman and Destin 2016; Kleebbua and Lindratanasirikul 2021). If the decision-making process is not set in an appropriate way, it can considerably disrupt creativity and its sustainability. Such an example was described in the study conducted by Malik et al. (2019). Deliberate decisions of the managerial staff on the working climate led to the unwillingness of academic members to share their knowledge. This significantly affected the overall creativity in a negative way. It is necessary to focus on creating an educational climate that encourages students to reach their full potential (Stephens et al. 2012; Smeding et al. 2013; Jury et al. 2015).

Based on original knowledge and findings from the area of building a high-quality educational environment, the positive impact of psychological safety and teachers' authentic guidance (Gunasekera et al. 2021; Price 2021) can be highlighted. By meeting the premise of open communication, teachers will improve students' understanding of the topic (Berkovich and Gueta 2020; Zhang et al. 2020; Corriveau 2020; Schnackenberg et al. 2021). These impacts create an environment that contributes to greater students' engagement (Soares and Lopes 2020).

A quality teaching process should include specific learning strategies and unique approaches, set via targeted decision-making by teachers and managers. The importance of an intentionally selected supportive approach is being underlined even outside the university environment, but still in connection to knowledge-intensive, intellectual types of work activity. Shafi et al. (2020) point towards a positive effect of transformational leadership and its individual dimensions on the environment members' creativity. Students will be supported at a high motivational level towards self-development and creativity, leading to more productive individuals (Elumalai et al. 2020; Abdukhalikova 2021; Torlak et al. 2021; Vlasova et al. 2017).

A wide range of specific opinions and findings was narrowed by the authors' analysis via the funnel effect (Appendix A). The result was therefore a confrontation of the preliminary taxonomy with other researchers' results. Subsequently, the final form of the taxonomy of factors influencing creativity of an organization's members was created (Appendix B).

## 4. Materials and Methods

The paper aims to create a taxonomy of decision-making factors that will affect the building of sustainable creativity of an organization's members. The application of this taxonomy will thus support the improvement of the working and study conditions. Based on the current knowledge as well as other already conducted research projects, three research hypotheses were established. These were listed in the subchapters of the theoretical background included in Section 1.

Following the surveys conducted thus far, the authors state that motivation and creativity are influenced by similar factors (Tumová and Blašková 2021). They focused their research on *the decision-making process,* which helps build high quality education.

The data used to test the validity of the hypotheses defined were obtained via the method of sociological inquiry, using two questionnaire surveys (25 and 21 questions). These focused on exploring the areas of motivation, creativity, and decision-making at universities. The respondents of the first survey were students, and in the second case, respondents were employees and managers at universities.

The number of students participating in the survey in 2019 was $n = 419$. There were 105 393 full-time university students in Slovakia in the same year (Statistical Office of the Slovak Republic 2019). With a tolerable error of 4.78%, this sample can be considered representative. The questionnaire focused on employees and managers in the academic environment was performed at a specific faculty at a university in Slovakia in 2019. The number of responses obtained ($n = 90$) in comparison with the total number of employees of the given faculty creates a representative sample with a tolerable error of 5.43%.

The mathematical-statistical analysis included the application of the Chi-Square Test of Independence and the Z-Score Test. These are suitable for identifying the dependencies

among the categorical variables studied. Other methods included deduction, induction, scientific abstraction, meta-analysis, and comparison.

## 5. Conclusions

This paper was devoted to the design of decision-making factors influencing the achievement of sustainable creativity in organizations. Therefore, the authors focused on three specific areas *(motivation, creativity, decision-making process)*. The justification of the focus on these areas is confirmed by the opinions of other scientists (Section 3). During the research, different perceptions of motivational programs at universities were revealed. Motivational programs are perceived much more by students than by employees. In the academic environment, individualized motivational programs are applied, especially towards students, and there are no group and organizational motivational programs focused on employees.

Therefore, it can be stated that the aspect of individuality (Appendix B) is currently being applied in the use of motivational programs. The discrepancy between the perception of students and staff also demonstrated in the utilization of creative methods. Teachers naturally support creativity in students, even without realizing it. Thus, the support of creativity belongs among their latent abilities. From the authors' perspective, it is important to focus not only on ongoing individualized support but also on support of creativity among other organization's members.

Another element in the taxonomy (Appendix B) is management. The importance of this element and the factors included in it are supported by the fact that the application of creative decision-making approaches truly influences the achievement of sustainable creativity (H2 hypothesis). The results of the surveys highlighted the importance of creative cooperation, included in the element of relationships. The last element was the atmosphere. It was justified both by the other scientists' opinions (Section 3) and the validity of the H3 hypothesis from the students' perspective.

Each of the elements contains examples of specific factors with their interpretation (Appendix B). The described parts form the core of the taxonomy. The left side then reflects the processes that are implemented using the elements. The first step is indirect support of creativity, which is an awareness of the current state of using motivation to support the creativity of the university members. Only then can direct support of creativity can follow. The right side presents the influences that will manifest in the academic environment. The application of taxonomy will bring the cultivation of decision-making, stimulate social action, and improve working lives.

Although the above-presented taxonomy reflects the views of authors and research teams from around the world, its final form is influenced by the results of a survey conducted in one Central European country. Therefore, taxonomy adapted for other countries should reflect other cultural, social, spatial, financial, and systemic conditions.

A peculiarity in considering sustainability, motivation, and creativity in terms of systematic quality improvement of higher education is that quality represents the qualitative phenomenon, identically to all other researched phenomena. The quality of these phenomena determines the resultative quality of higher education institutions. Vice versa, the sustained quality of universities determines the quality of sustainable motivation and creativity.

**Author Contributions:** Conceptualization: M.B., D.T. and M.M.; methodology: M.B., D.T. and M.M.; software: D.T. and M.M.; validation: M.B., D.T. and M.M.; formal analysis: M.B.; investigation: M.B., D.T. and M.M.; resources: M.B., D.T. and M.M.; data curation: D.T. and M.M.; writing—original draft preparation: M.B., D.T. and M.M.; writing—review and editing: M.B., D.T. and M.M.; visualization: D.T.; supervision: M.B.; project administration: M.B.; funding acquisition: M.B. and D.T. All authors have read and agreed to the published version of the manuscript.

**Funding:** The authors gratefully acknowledge the contribution of the Slovak Research and Development Agency under the project APVV-20-0481: Sustainability strategy of a sports organization in the

conditions of the Slovak Republic and the Research program of the Faculty of Security Management of Police Academy of Czech Republic in Prague under the project: Streamlining the functioning of the system of population protection and crisis management in the Czech Republic.

**Institutional Review Board Statement:** Not applicable.

**Informed Consent Statement:** Not applicable.

**Data Availability Statement:** Data is available on request from the authors.

**Conflicts of Interest:** The authors declare no conflict of interest.

## Appendix A

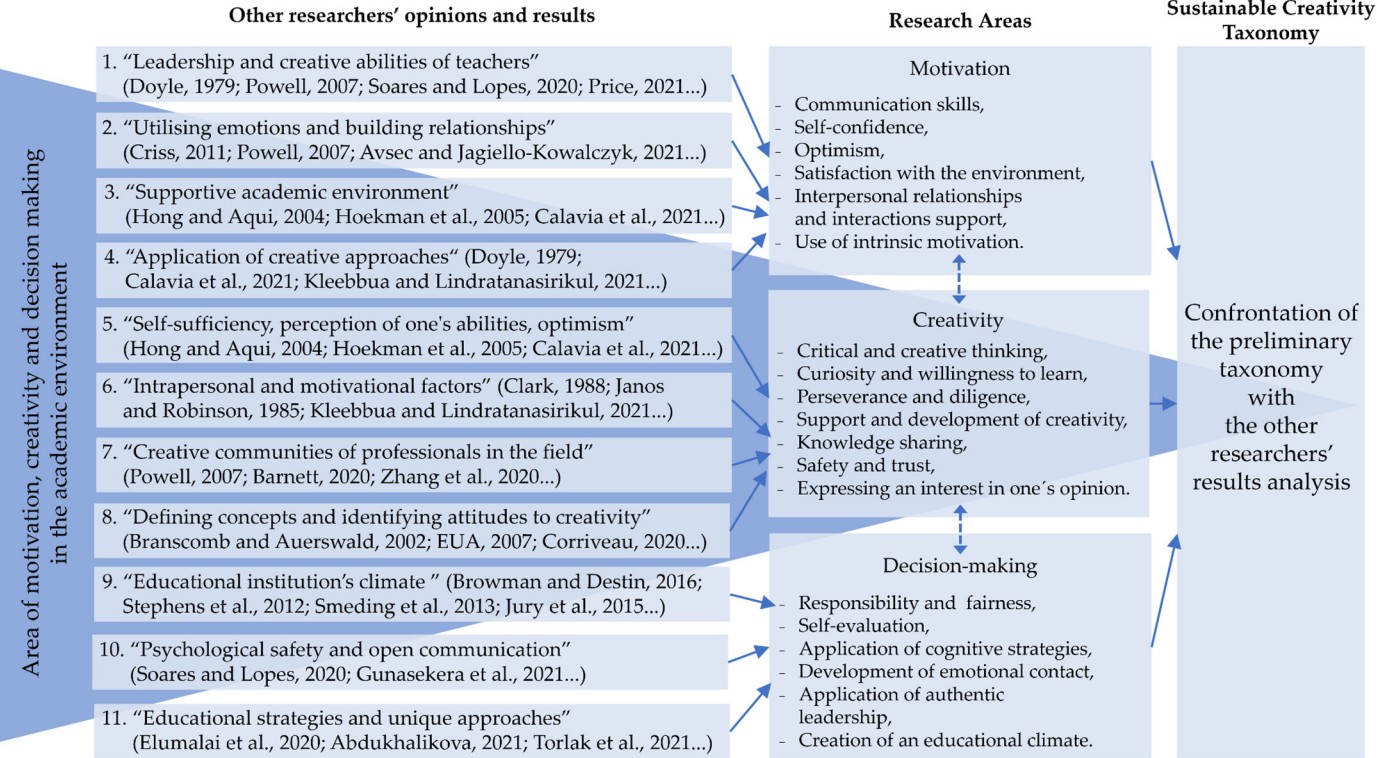

**Figure A1.** Other researchers' opinions in relation to sustainable creativity taxonomy.

**Appendix B**

**Figure A2.** Taxonomy of the factors affecting sustainable creativity.

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
