# Peer review of "Taxonomy of Factors Involved in Decision-Making to Sustain Organization Members’ Creativity"

_admsci, doi:10.3390/admsci12010039_

Round 1

Reviewer 1 Report

The work requires a review of the cited sources, some authors do not appear and errors are found, some of the figures used are not entirely legible. The selection of results is correct, although the comments could be expanded, as in the conclusions, there are current sources, but more relevant authors are missing on the subject used.

Author Response

Dear reviewer,

we deeply appreciate the time you spent carefully reading and reviewing our article. We apologize for the mistakes made in relation to the cited sources. We have thoroughly checked all the references in the whole text. We have added the missing sources and fixed other mistakes (e.g., incorrect years).

After consulting with the editor, we have decided to add the larger pictures as appendices, which considerably improved their legibility.

Based on another of your suggestions, we have added several relevant authors and their works (picking the most cited occurrences when searching for the pivotal keywords in the Web of Science database).

We believe that we have improved the quality of our article and that now it can be published in the journal.

Once again, thank you for your recommendations and the effort you put into your review.

Have a lovely day.

Authors

Reviewer 2 Report

This article entitled “Taxonomy of factors involved in the decision-making to sustain organization members’ creativity” is well written, uses adequate bibliographic references, the methodology is correct and provides novel results. It is a good text, very interesting. This study analyzes the quality of work and study conditions based on three processes in the academic world: motivation, creativity and decision-making aimed at supporting them in a Central European country. Although the results are interesting, as the authors themselves warn, in order to adapt these taxonomies to other countries, specific cultural, economic and systemic aspects should be taken into account. What would be missing, in my opinion, is to consider the sex of selected informants, to incorporate the gender perspective in the study, since I am sure that it would give us different results that we could interpret, depending on the gender roles and the sexual division of labor. of the universities studied. We understand that being a man or a woman directly influences the satisfaction of job expectations, the perception of risk in the workplace, etc. and it would be necessary to bear in mind the ideal representations about gender and work in the cultural context to more accurately interpret, for example, job satisfaction, creativity and motivation, both for teachers and students. It is a good, interesting article that can be published in the magazine as it is.

Author Response

Dear reviewer,

thank you for your kind words and all the praise you expressed about our article. We truly appreciate it. We have put a lot of work into writing this article, so we are really pleased by such positive feedback.

In relation to your suggestion on considering gender as an important factor, we will definitely incorporate it in our future research. Within this research project, we have already published some other results, even those analyzing the effect of gender, in other articles.

Once again, thank you for your recommendations and the effort you put into your review.

Have a lovely day.

Authors

Reviewer 3 Report

The manuscript presents a deep analysis of a valuable topic, but in this form, it cannot be recommended for publication. Scientific sources are added, the analysis methods seem to be correct. Based on a Trunitin test, it is an original work (similarity index is 7%). At the same time, the introduction, literature review and hypothesis are put together. The research question is not formulated. Methods are moved after the results. A minor issue is that the quality of the figures must be improved.

Please, follow the usual way of structuring a paper, and re-submit it.

Author Response

Dear reviewer,

first, let us kindly thank you for the time and energy you clearly had to put into your review. We were honestly pleased by all the positive points you listed in the review, commending on the depth of our analysis, originality supported by the low similarity index as well as the methods and sources we used.

Replying to your recommendations, we have added the research question’s formulation in the introduction section.

The larger pictures were added as appendices to make them much more readable (as this requirement was made by another reviewer too).

Regarding the article’s structure, while writing it, we were closely following the journal’s template. Based on your review, we have checked this with the editorial office. We believe that even though the article’s structure is not common it can be published in this form since it meets the journal’s formal criteria.

Thank you again for all your hard work put into your review.

Have a nice day.

Authors

Round 2

Reviewer 3 Report

I have checked the new version of the manuscript. According to the authors' reply, the changes made in the text, and the opinions of the other reviewers, I can accept the manuscript. I have no further requests.